# Extrusion fountains are hallmarks of chromosome organization emerging upon zygotic genome activation

Aleksandra Galitsyna [1,13] ✉, Sergey V. Ulianov[2,3,13], Mariia Bazarevich[4,5,6], Nikolai S. Bykov [7,12], Marina Veil[4], Meijiang Gao[4,5], Kristina Perevoschikova [8], Mikhail S. Gelfand [9], Sergey V. Razin [2,3], Leonid Mirny [1,10,14] ✉ & Daria Onichtchouk [4,5,11,14] ✉

The initiation of gene expression during development, known as zygotic genome activation (ZGA), is accompanied by massive changes in chromosome organization. However, the earliest events of chromosome folding and their functional roles remain unclear. Using Hi-C on zebrafish embryos, we discovered that chromosome folding begins early in development with the formation of fountains, distinct elements of chromosome organization. Emerging preferentially at enhancers, fountains show an initial accumulation of cohesin, which later redistributes to CTCF sites at TAD borders. Knockouts of pioneer transcription factors driving ZGA enhancers cause a specific loss of fountains, establishing a causal link between enhancer activation and fountain formation. Polymer simulations demonstrate that fountains may arise as sites of facilitated cohesin loading, requiring two-sided but desynchronized loop extrusion, potentially caused by cohesin collisions with obstacles or internal switching. Moreover, we detected cohesin-dependent fountain patterns at enhancers in mouse cells and found them reemerging with cohesin loading after mitosis. Altogether, fountains represent enhancer-specific elements of chromosome organization and suggest that chromosome folding during development and after cell division starts with facilitated cohesin loading. Observations in multiple systems further support facilitated loading at enhancers as a widespread phenomenon.

Zygotic genome activation (ZGA) is a critical point in the development of multicellular organisms. It is largely driven by the binding of pioneer transcription factors that open up chromatin and activate enhancers, leading to the awakening of zygotic transcription (reviewed in refs. 1,2). ZGA also coincides with dramatic reorganization of chromosome folding. In *Drosophila*, *Xenopus*, zebrafish, medaka, human, and mouse, the hallmarks of chromatin organization gradually emerge after ZGA, with largely featureless organization before ZGA[3-10]. As observed by Hi-C in all vertebrates, these hallmarks include compartments at larger scales, as well as topologically associated domains (TADs), dots and stripes, at a sub-megabase scale. Central to the formation of these sub-megabase structures is the process of loop extrusion by cohesin. Interacting with extrusion barriers such as CTCF[11,12], cohesin forms highly dynamic[13-15] chromatin structures. Functionally, cohesin-mediated loop extrusion and its modulation by CTCF play key roles in interactions between distal enhancers and their target promoters, and are crucial for development and differentiation[16-20].

Although the establishment of specific chromosome organization during development coincides with the major ZGA wave, the causal

relationship between these two processes remains unclear. In zebrafish, ZGA is driven by pioneering transcription factors (TFs) Pou5f3, Sox19b, and Nanog[21,22] (homologs of mouse pluripotency TFs[23]), which open chromatin, establish early enhancers, promote H3K27 acetylation, and activate transcription[24–26].

The initial steps of chromosome folding may also be guided by pioneer TFs that can load cohesin or recruit insulators, such as CTCF. For example, in *Drosophila*, the removal of the pioneer factor Zelda, a zygotic genome activator, leads to the weakening and loss of a subset of domain boundaries[3]. In zebrafish, cohesin and CTCF levels gradually increase after ZGA[27], but the factors and mechanisms responsible for their recruitment remain largely unknown. Overall, the mechanisms underlying this dramatic initial chromosome folding during development, and the roles of enhancers and pioneer TFs in this process, are still not fully understood.

Here, we generated Hi-C maps at several time points during early development for wild-type and mutant zebrafish embryos lacking various combinations of key pioneer transcription factors. Our analysis revealed that, upon ZGA, chromosomes begin to fold by forming "fountains," a distinct class of Hi-C features that we systematically detected and functionally characterized. Fountains emerging in early development resemble structures observed in quiescent mouse thymocytes ("jets")[28], Wapl-CTCF-depleted mouse cells ("plumes")[29], roundworm *Caenorhabditis elegans*[30,31], fungus *Fusarium graminearum*[32], plant *Arabidopsis thaliana*[33]; see Supplementary Note I "Fountain-like structures in other biological systems".

Surprisingly, fountains are enriched at early enhancers, rather than at active promoters or other functional genomic elements. Surrounded by genes active in early development, fountains transform into other elements of chromosome organization as development progresses. We hypothesized and demonstrated through simulations that these initial enhancer-specific structures are driven by facilitated cohesin loading at enhancers. TFs knockouts supported this mechanism, showing that disruption of enhancer activity leads to impaired fountain formation. Furthermore, we identified fountains as a collective signature of enhancers in mouse cells, suggesting that they represent a general enhancer-associated folding mechanism. Their cohesin-dependent nature is evident from their disappearance upon induced cohesin degradation, their absence during mitosis, and their gradual reemergence after mitosis. Consistently, fountain-like structures were also found to be cohesin-dependent in thymocytes[28], mouse embryonic stem cells (mESCs)[29], and *C. elegans*[30,31]. Overall, our findings indicate that chromosome folding during development is initiated by facilitated cohesin loading at enhancers, revealing the role for enhancers as active players in chromosome folding.

## Results

### Large-scale chromosome reorganization in early zebrafish embryogenesis

To characterize the dynamics of chromosome organization, we performed Hi-C for sperm and four stages of *Danio rerio* embryonic development: the last cell cycle before the major ZGA wave (2.75 hours post fertilization, hpf), late blastula (5.3 hpf, 2.3 hours after ZGA), 3-somite stage (11 hpf), and pharyngula (25 hpf)[34] (Fig. 1a).

While Hi-C maps were largely featureless before ZGA, the hallmarks of chromosome organization started to emerge at 5.3 hpf. At the global level, we noticed the formation of compartments and a strong Rabl configuration, i.e., enrichment of interactions between centromeric regions (Supplementary Fig. 1a, b). At the local level (<1 Mb), after ZGA (5.3 hpf), we observed the formation of features, referred to below as fountains (Fig. 1b–d). Fountains are distinct from TADs, yet morph into TADs at much later stages (Fig. 1b–e). At 11 hpf, compartments and TADs became more pronounced, and practically fully formed as compared to 24 hpf stage (Fig. 1e, and Supplementary Fig. 1b,c). While early Hi-C studies of zebrafish development suggested

presence of TADs and compartments before ZGA[35], later reports[4] as well as our data indicate absence of features in chromosome organization before ZGA (Fig. 1a–e), consistent with observations in other species[3–10]. Fountains appear to be the earliest features emerging after ZGA, suggesting that chromosome folding starts with the formation of fountains.

For comparison of our datasets with the published Hi-C data on *Danio rerio* development[4,35], we constructed developmental trajectories by two independent approaches: (i) pairwise correlation-based comparisons and embeddings of Hi-C maps into 2D space, and (ii) comparing datasets in their degrees of compartmentalization and insulation at TAD boundaries. Both approaches demonstrate that our datasets are well-aligned with the previous data. Interestingly, we observe that compartmentalization and insulation grow together, gradually and in synchrony with each other, during early development (Fig. 1f).

### Emergence of loop extrusion activity upon ZGA

Gradual increase in insulation (Fig. 1f, and Supplementary Fig. 1c) suggests the rising level of loop extrusion activity shortly after ZGA and its increase during embryonic development. Three lines of evidence support growing extrusion activity. (i) While TADs (quantified by the strength of insulation) remain weak at 5.3 hpf, they become more pronounced at later stages (Supplementary Fig. 1c). (ii) Analysis of the contact probability P(s) curves provides complementary evidence as they can reveal the presence of loops, irrespective of barrier elements. Extruded loops create a hump on the P(s) curve, best seen in its log-derivative[36,37]. We observe dramatically different slopes of P(s) before and after ZGA, with signatures of ~50 Kb loops starting to emerge as early as 5.3 hpf, yet becoming pronounced and larger (~100–150 Kb) at 11 hpf (Supplementary Fig. 1d). (iii) Published protein abundance data show presence of Rad21 cohesin component in the chromatin fraction as early as 5.3 hpf[27] and its growth with time. Together, this suggests that cohesin-driven loop extrusion is initiated at ZGA, and strengthens through development. Furthermore, this early loading of cohesin is consistent with our observation of fountains upon ZGA (Fig. 1b, c), if fountains are formed due to localized cohesin loading. Indeed, fountains (jets) in another system[28] were associated with cohesin loading. Functionally, early cohesin loading may be required for mediation of enhancer-promoter interactions[16–20].

### Fountains are transient structures emerging after ZGA

Fountains emerge at 5.3 hpf, likely representing early stages of chromosome folding. We define a fountain as a pattern of contacts that emanates from a single genomic locus and broadens with distance from the diagonal (Fig. 1b, c). To systematically characterize fountains, we developed an automatic *cooltools*-based[38] algorithm, *fontanka*, which is able to distinguish fountains from potential genomic misassemblies (Supplementary Fig. 1e, f, see "Methods" and Supplementary Methods "Fountain calling with *fontanka*"). Efficient fountain calling allowed us to see both individual patterns and their characteristic average shape: Fountains represent enrichments of contacts in an approximately 200 Kb range from the base, comparable in size yet very different in shape from TADs. In contrast to TADs, fountains are pinched near the diagonal, have a distinct base, and lack sharp boundaries (Fig. 1c,d).

Fountains are clearly distinct structures, different from TADs. Fountains do not correspond to either TAD centers nor TAD borders (Fig. 1d, Fig. 2a, and Supplementary Note II "Detailed characterization of zebrafish fountains"). They do not colocalize with CTCFs or sites of insulation (Fig. 2a, and Supplementary Fig. 2c) and look distinct from stripes that emanate from CTCF sites 45 degrees from the diagonal[39].

Remarkably, using *fontanka*, we detected 1460 fountains consistent across two Hi-C replicates at 5.3 hpf (Supplementary Dataset 1), many more than seen in adult mouse thymocytes (38, jets[28]). Several

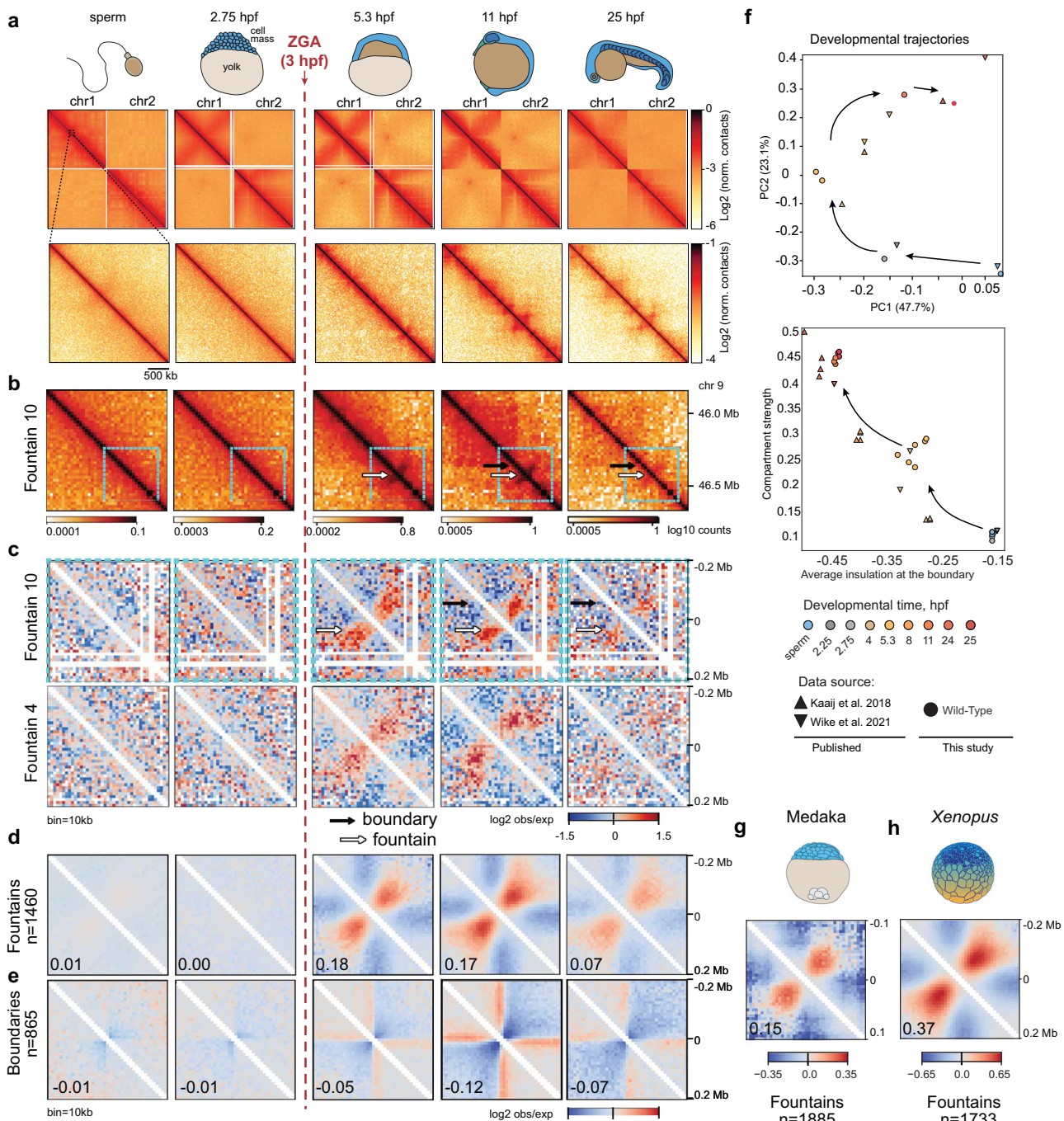

**Fig. 1 | Structural features of chromatin of embryogenesis and emergence of fountains after ZGA.** ZGA – zygotic genome activation; hpf – hours post fertilization; n – number of instances used for averaging; log2 norm. contacts – logarithm of Hi-C contacts normalized by iterative correction; log2 obs/exp – logarithm of observed Hi-C contacts divided by expected for each genomics separation; PC – principal component. Vertical dashed red line represents the ZGA onset in zebrafish development. Source data are provided as a Source Data file. **a** Representative examples of contact heatmaps showing dynamics of whole-chromosomal (top, chr1-2) and local contact patterns (bottom, chr10: 31,200,000-34,000,000). Dashed black line represents the scale of comparison between top and bottom. **b** Representative example of a fountain (white arrow) on the contact heatmap after ZGA (5.3 hpf). TAD boundary (black arrow) is detectable at 11 hpf. Region in blue frame is shown in detail in (**c**, top). Bin size = 20 Kb. **c** Examples of fountains in Hi-C maps normalized by expected. Top: the same region as in (**b**), bottom: another

example (see Supplementary Data for more examples). Bin size = 10 Kb. Coordinates represent the distance from the center of the Hi-C snippet. **d** Average fountain (defined at 5.3 hpf, n = 1460). Numbers represent the average fountain score. Coordinates represent the distance from the fountain base. **e** Average TAD boundary (defined at 11 hpf, n = 865). Numbers represent the average insulation score. Coordinates represent the distance from the boundary. **f** Developmental trajectories based on stratum-adjusted correlation coefficient (SCC, top) and average insulation plotted against compartmental strength (bottom). Hi-C datasets are from three studies ref. ([4,35], this work). Percentages label the variance explained by the PCA components. **g**, **h** Fountains at ZGA in other species. Fountains called using *fontanka*, reported in Supplementary Dataset 2. **g** Average fountain in Hi-C of medaka fish *Oryzias latipes* for developmental stage 10 (early blastula) from[6]. **h** Average fountain in Hi-C of frog *Xenopus tropicalis* for developmental stage 11 (gastrula) from[9].

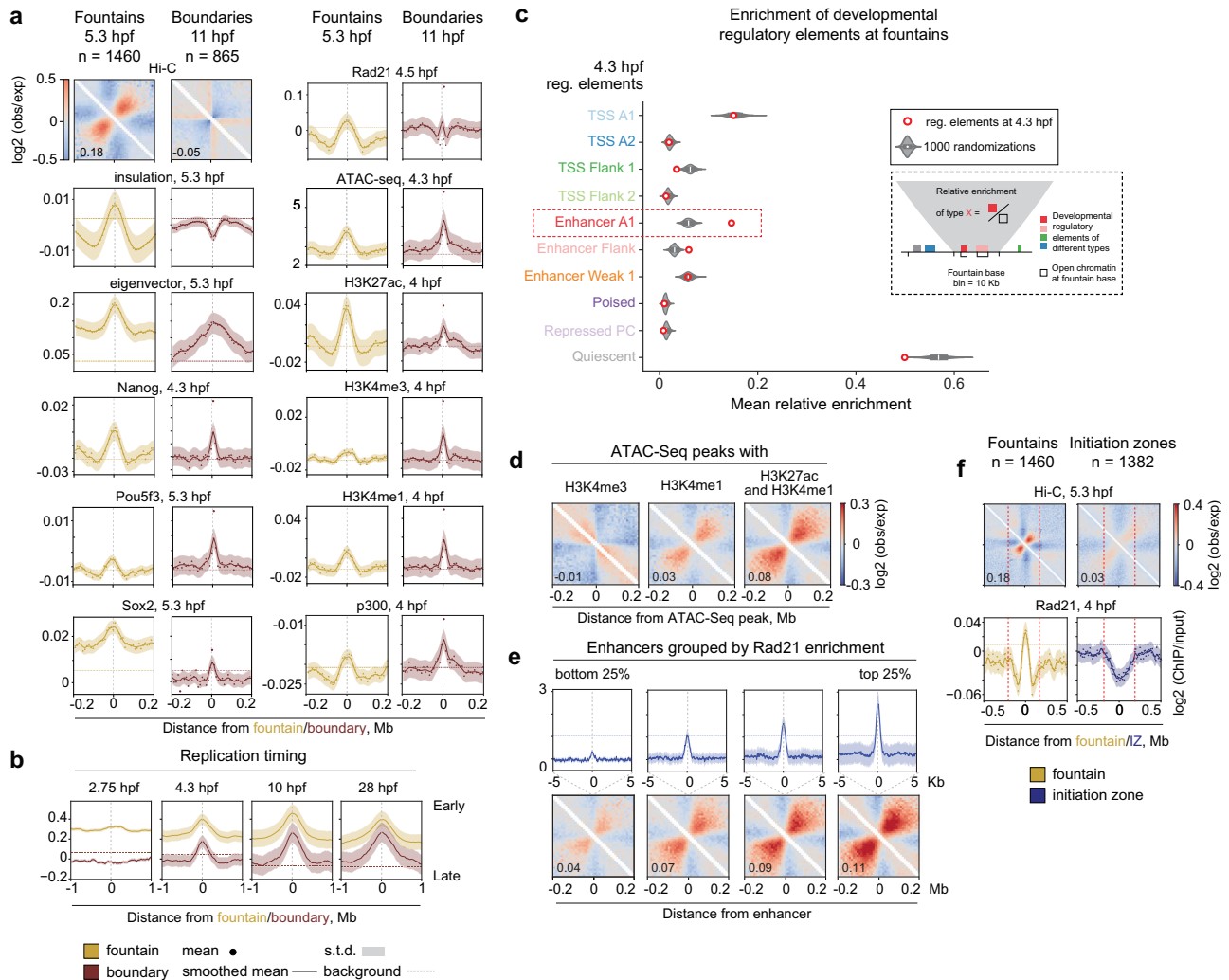

**Fig. 2 | Functional characterization of fountains.** Source data are provided as a Source Data file. **a** Features of fountain bases compared to TAD boundaries. Units: insulation score[38,83]; Nanog[84], Pou5f3[25], Sox2[22], p300[25], and histone modifications[25]: log2 ChIP-seq over input; ATAC-seq[42]: number of reads. **b** Fountain bases replicate earlier than TAD boundaries. Average replication time at indicated developmental stages (Repli-seq from[40]). **c** Fountains colocalize with enhancers at 4.3 hpf: relative enrichment of a regulatory element at the fountain is defined as a fraction of the genomic bin with the fountain base covered by this element. Dots: consensus predicted ATAC-seq-supported developmental regulatory elements (cPADREs, ATAC-seq peaks at the dome stage, annotated by 10 ChromHMM states) from[41]. Control violins: 1000 randomizations of fountain positions. Significantly enriched over control: Enhancer A1 (FDR«1e-4), Enhancer Flank (FDR«1e-4, based on normal distribution fit, see "Methods"). Boxes represent the quartiles of the fountain score distributions, the center line is the median, whiskers extend to 1.5 interquartile range. **d** Fountains form at enhancers but not at promoters. Average pileups of Hi-C maps at 5.3 hpf for regions enriched in promoter (H3K4me3) and enhancer (H3K4me1, H3K27ac) histone marks (see also Supplementary Fig. 3a–i). **e** Fountain strength at enhancers reflects cohesin occupancy. N = 5426 active enhancers (dome regulatory elements Enhancer A1 from[41]) were sorted into four groups by descending Rad21 ChIP-seq (from[4]). Top panel: cohesin occupancy profiles. Bottom panel: average pileups of the enhancers. Note that fountain strength goes up with increasing cohesin occupancy (see also Supplementary Fig. 2b). **f** Comparison of fountain bases (n = 1460) with replication Initiation Zones at 4.3 hpf (IZs, from[40]). Only IZs closest to the fountain bases were taken into account (n = 1382). Red dotted lines mark +/−0.2 Mb from the summit. (top) Hi-C signal, 5.3 hpf. Note that specific fountain structure forms at the fountain bases but not at IZs. (bottom) Rad21 occupancy, 4 hpf[4], around fountains versus replication IZs. Note that fountain bases are enriched in cohesin, while IZs are depleted of cohesin (see Supplementary Fig. 2d, i).

studies used our algorithm, *fontanka*, and found numerous fountains in *Caenorhabditis elegans*[30,31], growing *Arabidopsis thaliana*[33], and upon Wapl/CTCF-double depletion in mouse ES cells[29] (Supplementary Table 1 and Supplementary Note I). A much smaller number of fountains in adult mouse thymocytes[28] likely reflect biological differences, since algorithmically two approaches for fountain calling are similar (Supplementary Fig. 1g).

We next examined how fountains change through development. We observe that fountains gradually transform into other patterns at 25 hpf, such as dots and TADs, which are known to be formed by loop extrusion (Fig. 1b, c). Fountains are nevertheless visible at 11 and 25 hpf as an average pattern but hardly detectable individually, possibly

masked by TADs (see Supplementary Note III "Limitations of fountain detection as an average pileup"), interrupted by CTCFs (see also Supplementary Note IV "Realistic simulations of enhancer-targeted cohesin loading with CTCF barriers"), or suppressed by other mechanisms (Fig. 1d).

Next, we asked whether fountains are present during early development in other species. We analyzed Hi-C data for development of medaka fish *Oryzias latipes*[9] and frog *Xenopus tropicalis*[6]. Using our fountain caller, we identified individual fountains at stages close to ZGA in both species, revealing a clear fountain signal similar to that observed in zebrafish (Fig. 1g,h, Supplementary Dataset 2, Supplementary Note V "Fountain calling in medaka and *Xenopus*").

Together, this analysis shows that fountains constitute patterns distinct from known hallmarks of chromosome organization: TADs, stripes, and compartments. Fountains are not unique to zebrafish, as they are also visible in other externally developing early embryos of medaka fish and frog.

## Fountains are enriched at enhancers

Aiming to understand the functional role of fountains and their relationship to genome activation, we examined enrichment of genomic features at and around fountains. For comparison, we performed the same analysis for TAD boundaries that become visible at 11 hpf (Supplementary Dataset 3).

Both fountains and TAD boundaries were enriched in regions of open chromatin, active chromatin marks, cohesin, Pol II and pioneer TF binding as early as 4-5.3 hpf (Fig. 2a, and Supplementary Fig. 2a–c). Most of the fountains are located in A compartment at 5.3 hpf (65.6% of fountains versus 47.8% expected by chance), and are predisposed to avoid the future boundaries (0.2% versus 1.1% expected by chance). We also compared the replication timing between the fountain bases and boundaries, using the Repli-seq data at 2.75, 4.3, 10 and 28 hpf[40]. We found that, consistent with their euchromatic location, fountains were replicating significantly earlier than boundaries in all time points (Fig. 2b, Supplementary Fig. 2d).

However, fountains showed strong enrichment of enhancer-specific mark H3K4me1 and H3K27ac, and rather weak enrichment of the promoter-specific H3K4me3 mark. TAD boundaries, in contrast, did not display this preference for enhancer marks and were about equally enriched in H3K4me1 and H3K4me3 (Fig. 2a, also Supplementary Note II). Together, this suggests that fountains are enriched in enhancer marks and depleted in promoter marks showing patterns distinct from that of TAD boundaries.

To further characterize the functional states of fountains, we examined the enrichment of developmental regulatory elements at 4.3 hpf (ATAC-seq peaks annotated by ChromHMM chromatin states in development[41]). Such functional categories include promoters of different levels of activity, active enhancers, Polycomb repressed promoters, heterochromatin, and quiescent chromatin. Surprisingly, we found that among ten developmental regulatory element types, only the enhancer states show significant enrichment at fountains, as compared to a random control (FDR«1e-4 based on normal distribution fit, Fig. 2c). Neither transcriptionally active states, such as transcription start sites, nor inactive or repressed states are enriched at fountains as compared to a randomized control (see Supplementary Methods "Enrichment of developmental regulatory elements at fountains"). We found that 854 fountains are associated with sites of active transcription (ATAC-seq peaks with marks of enhancers or promoters, located within 10 Kb distance from the fountain bin, with 670 expected in the random control), with 625 fountains containing at least one enhancer (with 342 expected in the random control).

As enhancers tend to be located close to their target promoters, we asked whether fountains were proximal to active promoters. We found that fountains were indeed significantly enriched in the proximity of zygotic genes that began transcription between 3 and 5.5 hpf ($p$ value ~ $10^{-8}$, Supplementary Fig. 2f, see Supplementary Note II "Detailed characterization of zebrafish fountains"), with about ~50% of fountains having an active gene less than 200 Kb away; and ~70% (1042) fountains containing promoters active at dome stage within a 50 Kb window.

To further explore connection between fountains and enhancers, we performed a complementary analysis that does not rely on a priori identification of fountains. We stratified all ATAC-seq peaks by their enhancer (H3K4me1) and promoter (H3K4me3) marks and compared average Hi-C pileups centered at ATAC-seq peaks from different categories (Fig. 2d, and Supplementary Fig. 3a, b). We observed striking differences in Hi-C patterns on promoters and enhancers. Enhancers showed a pronounced fountain pattern that became even

stronger when active enhancers were selected using a combination of high H3K4me1 and high H3K27ac (Fig. 2d, and Supplementary Fig. 3d, e, control: Supplementary Fig. 3g, h). On the contrary, Hi-C pileups centered at promoters showed a pattern opposite to that of a fountain, i.e., an average insulation (Fig. 2d, detailed analysis in Supplementary Fig. 3). This analysis, independent of fountain calling, complements the observation of enhancer enrichment at fountains and demonstrates that fountains form preferentially at enhancers rather than promoters.

Our developmental timeline allows us to examine the relationship between fountains formed upon ZGA (5.3 hpf) and enhancers activated later in development (12 hpf). We found that 5.3 hpf fountains are also enriched in 12 hpf enhancers (Supplementary Fig. 2e). Moreover, when we considered later enhancers alongside early ones, 765 fountains contained at least one such enhancer (versus 625 containing 5.3 hpf enhancers). This suggests that at least some fountains are formed at sites that will establish their enhancer status later in development.

## Fountains form at cohesin-enriched sites within early replicating regions

In zebrafish, the DNA replication timing program is progressively refined in parallel with the lengthening of the cell cycle in development[40]. As fountains and non-random replication initiation zones (IZs) are established within the same developmental time window, we inquired whether fountains mark the early replication forks initiating on IZs (Supplementary Dataset 4). However, the averaged Hi-C map of IZs[40] does not resemble the average fountain (Fig. 2f), and the fountain strength does not depend on the distance to IZs (Supplementary Fig. 2i). Aiming to investigate the possible relationship of fountains to replication, we compared the genomic properties of fountain bases and IZs. Both IZs and fountain bases were enriched in open chromatin and enhancer marks (Supplementary Fig. 2d), consistent with the location of fountains in active (and hence early-replicating) chromatin. However, strikingly, fountain bases were enriched in cohesin, while IZs were rather depleted of it (Fig. 2f, and Supplementary Fig. 2d). Taken together, this suggested that cohesin accumulation on enhancers results in fountain formation independently of replication initiation.

## Cohesin accumulation is associated with fountain strength

Overall, one of the most distinct features of fountains (Fig. 2) was the accumulation of cohesin. We examined this connection by two approaches, one that uses called fountains and the other that does not. Among identified fountains, those with higher cohesin (Rad21 subunit) levels showed a stronger fountain pattern (Supplementary Fig. 2b). Independent of fountain calling, we grouped strong enhancers by cohesin levels and found that higher cohesin signal correlated with a stronger fountain pattern (Fig. 2e). Together, this indicates that fountains preferentially form at cohesin-rich enhancers. Perturbation experiments below allow us to test a causal connection between enhancer activation and the establishment of fountains.

## Pioneer factors establish open chromatin platforms for fountain formation

Early establishment of enhancers in zebrafish is controlled by the binding of the key zygotic genome activators, transcription factors Pou5f3, Sox19b, and Nanog (PSN). These factors act as pioneers by creating open chromatin regions and recruiting histone modifiers, such as p300, to their binding sites[21,22,24–26,42]. If fountain formation requires chromatin accessibility and enhancer activity, we expect the removal of Pou5f3, Sox19b, or Nanog to disrupt fountain formation on the regions where these factors bind and open chromatin. To test this mechanism, we performed Hi-C on the 5.3 hpf embryos, null-mutant for Pou5f3[43], Sox19b[42], Nanog[44], and also on double or triple mutant embryos bearing combinations of two or three of these mutations (MZ*triple*) (Fig. 3a).

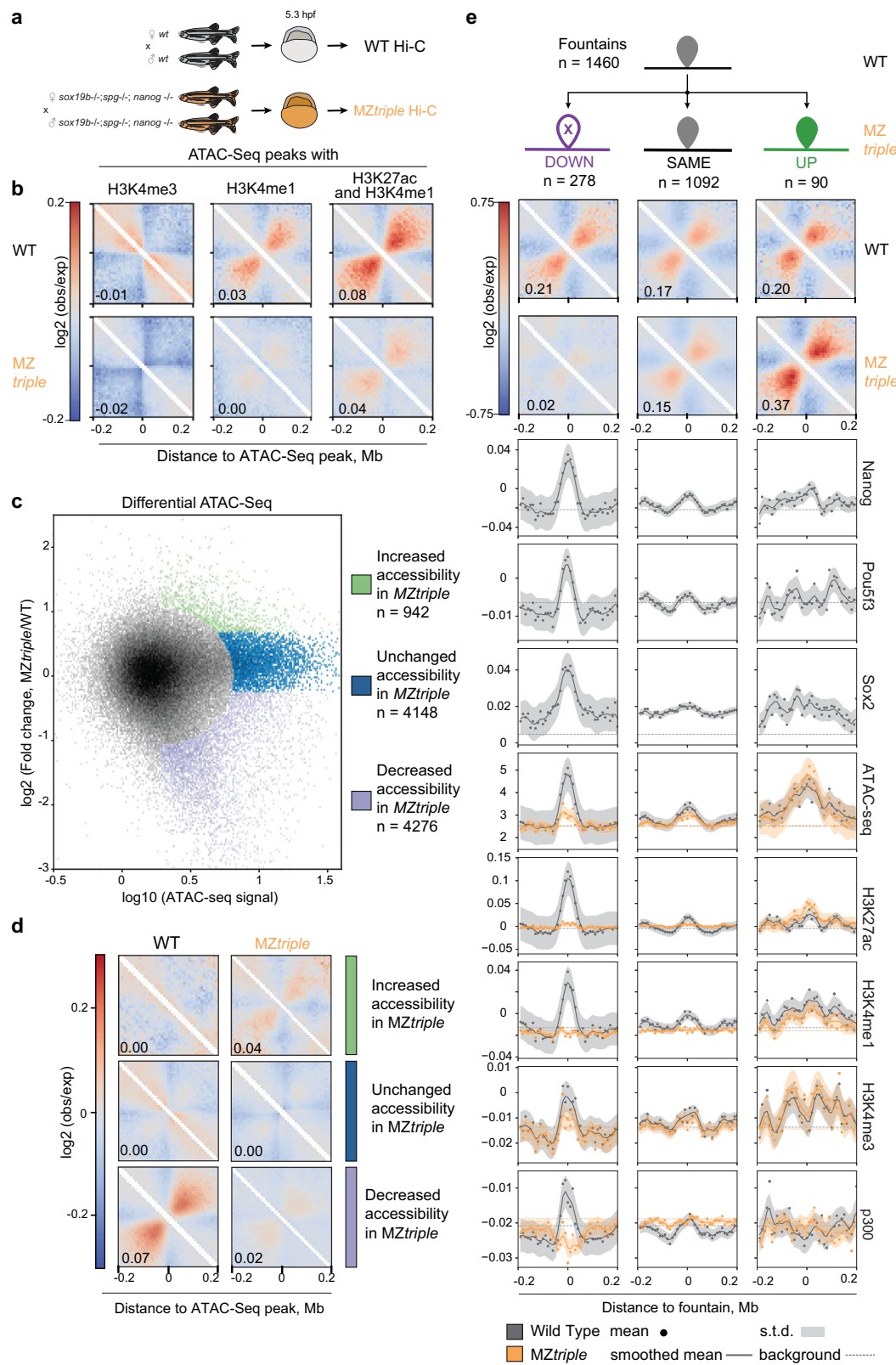

First, we examined changes in Hi-C patterns at four classes of ATAC-seq sites: enhancers, promoters, both, and neither (Fig. 3b). The fountain pattern, seen most distinctly for the enhancer group, has vanished in MZ*triple*, strongly supporting the role of enhancers in establishing fountains.

Next, we examined sites of open chromatin (ATAC-seq peaks) that changed in MZ*triple* (Fig. 3c). We found that sites that lost their accessibility in MZ*triple* showed a fountain pattern in the wild-type that vanished in MZ*triple* (Fig. 3d, and Supplementary Note II). Similar effects were seen for accessible sites with either H3K27ac or H3K4me1 marks (Supplementary Fig. 3j-k).

Finally, we performed a differential analysis of the identified fountains (Fig. 3e) and categorized them into three groups: fountains that weakened in MZ*triple* (Down, $n = 278$), remained unchanged

**Fig. 3 | Pioneer factors establish open chromatin platforms for fountain formation (three approaches).** Source data are provided as a Source Data file. WT – wild type; MZ*triple* – Maternal-Zygotic (MZ) homozygous Pou5f3/Nanog/Sox19b mutant. Numbers represent the average fountain strength for the pile of snippets. **a** Schematic of obtaining triple Pou5f3/Sox19b/Nanog zebrafish mutants to test the formation of fountains. See "Methods" and Riesle et al. 2023[45] for details. **b** Loss of fountain signature in MZ*triple* at the ATAC-seq peaks enriched with different types of active histone marks, similar to Fig. 2d. See Supplementary Fig. 3a–i for detailed explanation of the stratification design. **c**, **d** Chromatin openness in MZ*triple* mutant is coupled with fountain pattern formation in Hi-C. See also Supplementary Fig. 3j-h for a similar analysis of coupling of H3K27ac and H3K4me1. c. Distribution of ATAC-seq responses in MZ*triple* compared to wild type and definition of the groups of genomic bins based on their ATAC-seq responses. OX axis: logarithm of averaged ATAC-seq between wild-type and MZ*triple* for each genomic bin. OY axis: logarithm of fold change between wild-type and MZ*triple*. Data from[42,45]. **d** Average Hi-C pileup at ATAC-seq peaks from different groups from (**c**), wild-type and MZ*triple* at 5.3 hpf. **e** Weakened fountains in MZ*triple* mutant bear the strongest early enhancer features. From top to bottom: (top) Three groups of fountains by the change in fountain strength in MZ*triple*. Out of 1460 fountains, 278 weakened (DOWN), 90 strengthened (UP), and 1092 remained unchanged (SAME, see Methods "Differential fountains in MZ*triple*"). (middle) Pileups of wild-type and MZ*triple* Hi-C at DOWN-, SAME-, and UP-fountains. (bottom) Enrichment profiles of various features of chromatin of three groups of fountains. ATAC-seq: mean read count; other profiles: log2(ChIP-seq/input). ChIP-seq for Nanog at 4.3 hpf from[84] and for Pou5f3 and Sox19b at 5.3 hpf from[25], ATAC-seq data from[25]. H3K27ac and p300 ChIP-seq data for 4 hpf from[25].

(Unchanged, $n = 1092$), and became stronger in MZ*triple* (Up, $n = 90$). The weakened fountains showed the strongest enhancer signature and were the most enriched for pioneer transcription factors binding (PSN, Fig. 3e).

However, many fountains formed at this stage remain unaffected, suggesting that they are controlled by PSN-independent enhancers or by enhancers that become active at later stages of development. Below we show some of these PSN-independent enhancers become fully active later in normal development, accompanied by the strengthening of fountains at them.

Together, this triple-knockout has revealed that PSN-dependent enhancers active at ZGA form fountains. Moreover, abolition of their enhancer activity led to the loss of the fountain signature, indicating that enhancer activity is required for fountain formation at these sites.

Finally, we demonstrate that even individual pioneer TFs contribute to fountain formation as seen from changes in fountain strength for individual Pou5f3[43] and Nanog[44] mutants. Recently we found that Pou5f3 and Nanog promote chromatin accessibility in different genomic locations[45], and, thus, we used chromatin accessibility as a proxy for the Pou5f3-, Nanog- and Pou5f3/Nanog-dependent enhancers. In total, 441 out of 1460 fountain bases lose their accessibility in any of the mutants (30% of all fountains with FDR < 5%, Supplementary dataset 5). More thorough classification of these fountains revealed that TF-specific loss of chromatin accessibility led to specific loss of fountains (Supplementary Fig. 4). This analysis further strengthens the causal link between establishment of accessible chromatin by pioneer TFs and formation of fountains.

## Fountains are formed at functional sites that are established later in development

Since most of the fountains remained unchanged and some even got stronger in MZ*triple*, we asked next what distinguishes these fountains from the PSN-dependent ones that weaken in the MZ*triple* mutant. Indeed, fountains that strengthen in MZ*triple* also got stronger later in normal development, after gastrulation (11 hpf, Fig. 4a). They also became earlier replicating after gastrulation (Fig. 4b), and acquired H3K27ac after gastrulation (12 hpf, Fig. 4c). These observations echoed our recent finding[45] that late enhancers become aberrantly open and activated at ZGA in MZ*triple*.

Interestingly, the reduction of H3K27ac upon p300 inhibition or blocking of the RNApolII by flavoperidol at 4 hpf[4] resulted in some weakening of fountains but not in their collapse (Fig. 4d). This observation suggests that key to fountain formation are processes upstream of transcription initiation and H3K27 acetylation, such as TF binding and the establishment of accessible chromatin on enhancers.

The majority of fountains (~75%) that were unchanged in MZ*triple*, and thus are PSN-independent, may nevertheless be enriched for weaker functional sites and/or enhancers that get activated later in development. PSN-independent fountains were enriched in active enhancer marks, but less than the fountains of two other categories (Fig. 4c). Using functional annotation of regulatory sites[46], we found

that PSN-independent fountains are enriched in general developmental regulatory categories (Fig. 4e). Similarly to PSN-dependent groups, PSN-independent fountains were enriched in overlapping tissue-specific adult enhancers[47] (Fig. 4f).

Consistently, fountains that go down in MZ*triple* correspond to early active enhancers, i.e., had ATAC-seq peaks with active enhancer signature at 4.3 hpf (Fig. 4g). Fountains that go up in MZ*triple* correspond to enhancers active after gastrulation, i.e., were enriched with active enhancer signature at 12 hpf (Fig. 4h). Altogether, our analysis established a causal relationship between the changes of chromatin accessibility on enhancers and fountain formation.

## Facilitated loading of loop extruding SMCs as a mechanism of fountain formation

We argue that, while distinct from TADs and stripes, fountains reflect early chromosome folding by cohesin-mediated loop extrusion. First, we found that extruded loops of ~50 Kb emerge as early as 5.3 hpf (Supplementary Fig. 1d). Second, we observed cohesin accumulation at fountains at this stage (Fig. 2a, Supplementary Fig. 2a–c). Finally, fountains with high cohesin levels show a more pronounced pattern of contacts (Supplementary Fig. 2b).

Known extrusion-mediated patterns are formed by stopping loop extrusion at specific positions, resulting in the accumulation of cohesin at these positions[39,48]. Fountains are distinct from TADs and stripes yet also accumulate cohesin at their bases (Fig. 2a, and Supplementary Fig. 2a–c). Since we see no enrichment of CTCF at fountains (Supplementary Fig. 2c), cohesin accumulation there can be due to another mechanism. In fact, the strength of the fountain shape demonstrates no dependence on the distance to extrusion barriers, such as CTCFs and actively transcribing genes (Supplementary Fig. 2g–i). Thus, we hypothesize that cohesin accumulation at fountain bases, and fountains themselves, emerge due to facilitated cohesin loading.

To test the facilitated loading mechanism, we developed polymer simulations in which cohesin loads preferentially at specific sites (Fig. 5a). We defined the site of facilitated cohesin loading as a 1-Kb region (one simulation monomer), consistent with recent characterizations of regulatory elements[49] and the average length of ATAC-seq peaks at fountain bases (606 bp)[41], which corresponds to the narrow (~10 Kb) bases of the fountains we observe. Once loaded (facilitated or uniform), cohesins perform two-sided loop extrusion, stopping when they encounter one another[12].

## Fountain shape helps elucidate loop extrusion mechanism

Next, we identified conditions and mechanisms that lead to the formation of fountains in simulations. We observed that the loading of extruders only at the loading site and extrusion from there yields not a fountain, but rather a narrow hairpin pattern similar to that observed in bacteria (Supplementary Fig. 5a–c). Gradual broadening of experimental fountains (Fig. 5b) as they emanate from the fountain bases suggests that the extrusion path of cohesin is not perfectly symmetric, i.e., reeling of DNA on one side of cohesin is not perfectly synchronized

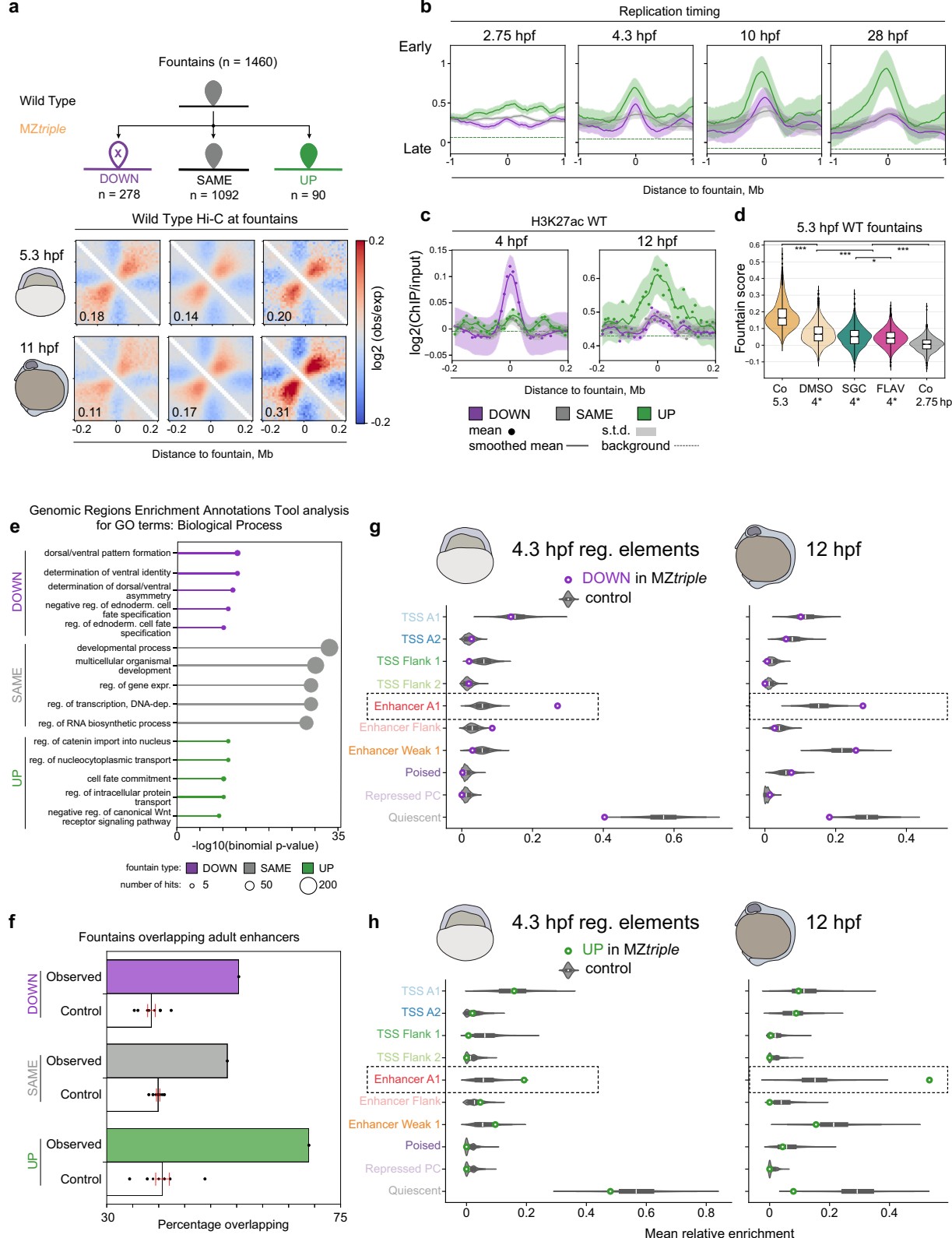

**a** Fountains (n = 1460)

Wild Type

MZ*triple*

DOWN n = 278 · SAME n = 1092 · UP n = 90

Wild Type Hi-C at fountains

5.3 hpf: 0.18, 0.14, 0.20

11 hpf: 0.11, 0.17, 0.31

Distance to fountain, Mb

log2 (obs/exp)

**b** Replication timing — 2.75 hpf, 4.3 hpf, 10 hpf, 28 hpf. Distance to fountain, Mb

**c** H3K27ac WT — 4 hpf, 12 hpf. log2(ChIP/input). Distance to fountain, Mb

DOWN · SAME · UP
mean ● · s.t.d.
smoothed mean — · background ·····

**d** 5.3 WT fountains. Fountain score. Co 5.3, DMSO 4*, SGC 4*, FLAV 4*, Co 2.75 hpf

**e** Genomic Regions Enrichment Annotations Tool analysis for GO terms: Biological Process

DOWN:
- dorsal/ventral pattern formation
- determination of ventral identity
- determination of dorsal/ventral asymmetry
- negative reg. of ednoderm. cell fate specification
- reg. of ednoderm. cell fate specification

SAME:
- developmental process
- multicellular organismal development
- reg. of gene expr.
- reg. of transcription, DNA-dep.
- reg. of RNA biosynthetic process

UP:
- reg. of catenin import into nucleus
- reg. of nucleocytoplasmic transport
- cell fate commitment
- reg. of intracellular protein transport
- negative reg. of canonical Wnt receptor signaling pathway

-log10(binomial p-value)

fountain type: DOWN · SAME · UP
number of hits: 5 · 50 · 200

**f** Fountains overlapping adult enhancers. DOWN: Observed/Control; SAME: Observed/Control; UP: Observed/Control. Percentage overlapping

**g** 4.3 hpf reg. elements / 12 hpf. DOWN in MZ*triple* · control. TSS A1, TSS A2, TSS Flank 1, TSS Flank 2, Enhancer A1, Enhancer Flank, Enhancer Weak 1, Poised, Repressed PC, Quiescent

**h** 4.3 hpf reg. elements / 12 hpf. UP in MZ*triple* · control. TSS A1, TSS A2, TSS Flank 1, TSS Flank 2, Enhancer A1, Enhancer Flank, Enhancer Weak 1, Poised, Repressed PC, Quiescent. Mean relative enrichment

with the other side. Asymmetric extrusion by cohesin, when averaged over cells, could produce symmetric fountains with increasing spread with distance from the loading site, as we measured in Hi-C (Fig. 5c, Supplementary Fig. 5a-c).

We suggest three mechanisms that can lead to such asymmetry in extrusion dynamics (Fig. 5c). The first mechanism is the intrinsic asymmetry of extrusion by cohesin, i.e., one-sided extrusion with direction switching, as was long anticipated[39] and recently observed in vitro[50]. The second mechanism is external to the motor and constitutes pausing of enhancer-loaded cohesins on randomly positioned barriers: while stochastically pausing at a barrier, cohesin can continue reeling chromatin on the other side. Such randomly positioned extrusion barriers can be molecules of MCMs[51] or transcribed genes[48]. The third mechanism is the pausing of enhancer-loaded cohesins at other

**Fig. 4 | Changes in enhancer repertoire underlie fountain changes in MZ*triple* mutant.** Source data are provided as a Source Data file. **a** Fountains upregulated in MZ*triple* mutant at 5.3 hpf ("UP") increase later in normal development. Top: three groups of fountains by regulation in MZ*triple*. Bottom: Hi-C average fountains at 5.3 hpf (WT2, two replicates) and at 11 hpf. **b** Replication timing for three groups of fountains ("UP", "SAME", DOWN") during developmental progression. **c** H3K27ac for three groups of fountains ("UP", "SAME", DOWN") at 4 hpf and 12 hpf. **d** Fountain strength is reduced in the embryos treated by p300 and RNApolII inhibitors (datasets marked by 4* hpf from[4]). Violin plots show fountain scores for 1460 fountains for the indicated conditions: DMSO control; SGC – p300 inhibitor; FLAV – RNApolII inhibitor. Co – this work. *P*-values of Tukey-Kramer test (see Source Data, p-value levels: *** – <0.001, ** – <0.01, * – <0.05). Boxes represent the quartiles of the fountain score distributions, the center line is the median, whiskers extend to 1.5 interquartile range, and outliers are shown as dots. **e** Top 5 Genomic Regions Enrichment Annotations Tool (GREAT[85]) enrichment categories in GO: Biological Process for "DOWN", SAME" and "UP" fountains. Dot size represents the number of hits, and OX position reflects the p-value of the term enrichment (binomial GREAT test, see Source Data). **f** Percentage of fountains overlapping enhancers from adult zebrafish tissues (data from[47]). Control: randomized fountain positions. Fountain (or control bin) was scored if it overlapped at least one enhancer for at least one nucleotide. Bin size: 10 Kb, error bars represent standard deviation, center line represents mean. N = 9 control samplings are shown as overlaid dots. **g** Fountains weakened in MZ*triple* mutant (DOWN) colocalize with early (4.3 hpf) enhancers. Developmental regulatory elements from[41], similar to Fig. 2c. Boxes represent the quartiles of the fountain score distributions, the center line is the median, whiskers extend to 1.5 interquartile range. **h** Fountains strengthened in MZ*triple* mutant (UP) colocalize with late (12 hpf) enhancers. Developmental regulatory elements from[41], analysis and elements notation similar to Fig. 2c.

cohesins. Such mechanism stipulates that cohesin loads uniformly along the genome, yet with a preferential loading at enhancers. This mechanism can also explain the weakening of fountains later in development due to the rising levels of cohesin on DNA[27].

For every model, we swept its parameters and selected models that best fit the shapes of fountains from Hi-C for 5.3 hpf (Fig. 5c, and Supplementary Fig. 5,6). All models have a range of parameters where they could closely approximate experimental fountains. For the model of background and facilitated loading, our parameter sweeps (Fig. 5d, e, and Supplementary Fig. 5a–c) revealed that ~10-fold facilitation at narrow (1 Kb) enhancer sites is sufficient to match the intensity profile of fountains closely. As for the background cohesins, the density of 1 cohesin per 100–200 Kb is required, consistent with cohesin density in other vertebrate systems[12]. Simulations of this mechanism reproduce the correlation of the fountain shape (Fig. 5e), intensity of signal at fountains (measured by protractor tool, Fig. 5d) and cross-sections of fountains at the genomic distances up to 200 Kb (Supplementary Fig. 5c). Finally, simulations also reproduce a broad (~50–100 Kb) peak of cohesin accumulation seen in ChIP-seq (Fig. 5b).

We also considered and ruled out an alternative mechanism where fountains are formed by affinities of fountain-proximal regions, i.e., compartment-like mechanisms. Such a mechanism would result in enrichment of contacts between fountains that we don't see in the data (Supplementary Fig. 5d).

Importantly, all extrusion-based mechanisms do not rely on CTCF for the formation or broadening of fountains, consistent with the absence of CTCF near fountains in zebrafish (Supplementary Fig. 2c). Consistently, a recent study by Lüthi et al.[30] observed prominent fountains in *C. elegans* that lack CTCF (lost in nematode evolution). This contrasts with the formation of stripes at CTCF-bound sites across vertebrate genomes[12,52]. Interestingly, in simulations, each fountain contains, on average, ~1 cohesin in the steady state. This indicates that they are different from bacterial hairpins, where multiple (~15-30) extruders of bacterial SMC were simultaneously moving along a hairpin from the loading to the unloading sites[53,54]. Together, our modeling suggests that fountains can be formed by one of these mechanisms or their combination: facilitated loading, interactions between cohesin and other extrusion obstacles, or internally desynchronized extrusion by cohesin motors.

### Cohesin-dependent fountain-like structures emerge at enhancers in mouse cells

To elucidate further the role of cohesin in generating fountains, we asked whether fountains are present in early development of mammalian species, and if their formation depends on cohesin. Uniquely, mouse systems allow to study the effect of cohesin loss due to its experimental depletion as well as natural cohesin loss during metaphase and its reloading upon entrance into G1.

In mouse embryonic stem cells (mESC), a model for the inner cell mass of the mouse preimplantation blastocyst, pioneer factors Pou5f1

and Sox2 mediate cohesin binding on enhancers[55–57], in striking parallel to our zebrafish system. We examined Micro-C from mESC, as well as from cells with acute cohesin, CTCF, and Wapl depletions[58]. Although we could not automatically call individual fountains as TADs, dots, and stripes overshadowed them, we piled Micro-C maps at CTCF-distal enhancers and promoters. In the control condition, we observed a distinct fountain pattern on enhancers but not on promoters, as we observed in fish (Fig. 6a, top, compare to Fig. 2d). Strikingly, the enhancer-associated fountain pattern vanished upon cohesin degradation (Fig. 6a, bottom).

We also observe CTCF-dependent changes in fountain shape (Fig. 6b, bottom), providing additional support that cohesin is the extruder underlying fountain formation. In untreated cells, the enhancer-centered fountain is compact, whereas upon CTCF degradation it becomes more extended and weak. To test whether these shapes of fountains emerged due to interactions between enhancer-loaded cohesins and CTCFs, we simulated 100 genomic regions with their specific CTCF arrangements around the enhancer (Supplementary Fig. 8b). Using parameters found for simulations in zebrafish, we see that CTCFs, when strong, indeed distort shapes of individual fountains, while changing the shape of an average fountain in the same way we observe in the experimental data (see Supplementary Note IV, "Realistic simulations of enhancer-targeted cohesin loading with CTCF barriers").

Moreover, Wapl depletion that leads to increased cohesin density and processivity shows much stronger fountains. Taken together, this analysis establishes a strong causal link between cohesin-driven extrusion and the formation of fountains at enhancers during the early stages of embryonic development. The cohesin-dependent nature of fountains was also established in thymocytes[28] and Wapl/CTCF-depleted cells[29], further supporting our conclusions.

If fountains are formed due to extrusion by cohesin, we expect them to vanish during mitosis and gradually reappear as cohesin reloads on chromosomes in early G1. To this end, we examined fountains in Hi-C data from cell cycle-synchronized populations of cultured G1E erythroblast cells[59]. Consistent with cohesin's role, fountains on piled enhancers were absent during prometaphase and metaphase, and began to reappear upon exit from mitosis, gradually strengthening from early to mid to late G1 phase (Fig. 6c). Importantly, we were able to observe individual fountains in early G1 (Fig. 6d, Supplementary Fig. 7), and see how they transform at later G1, morphing into TADs and stripes. It is probably the low level of cohesin at early G1 that makes fountains discernible[60]. This makes a striking parallel between ZGA fountains in zebrafish and early G1 fountains in mice – both appearing upon gradual loading of cohesin – in development and through the cell cycle. In both systems, the first events in chromosome folding are the loading of cohesin at enhancers.

## Discussion

By analyzing chromosome organization in early zebrafish development, we found that loop extrusion activity initiates upon ZGA. Yet

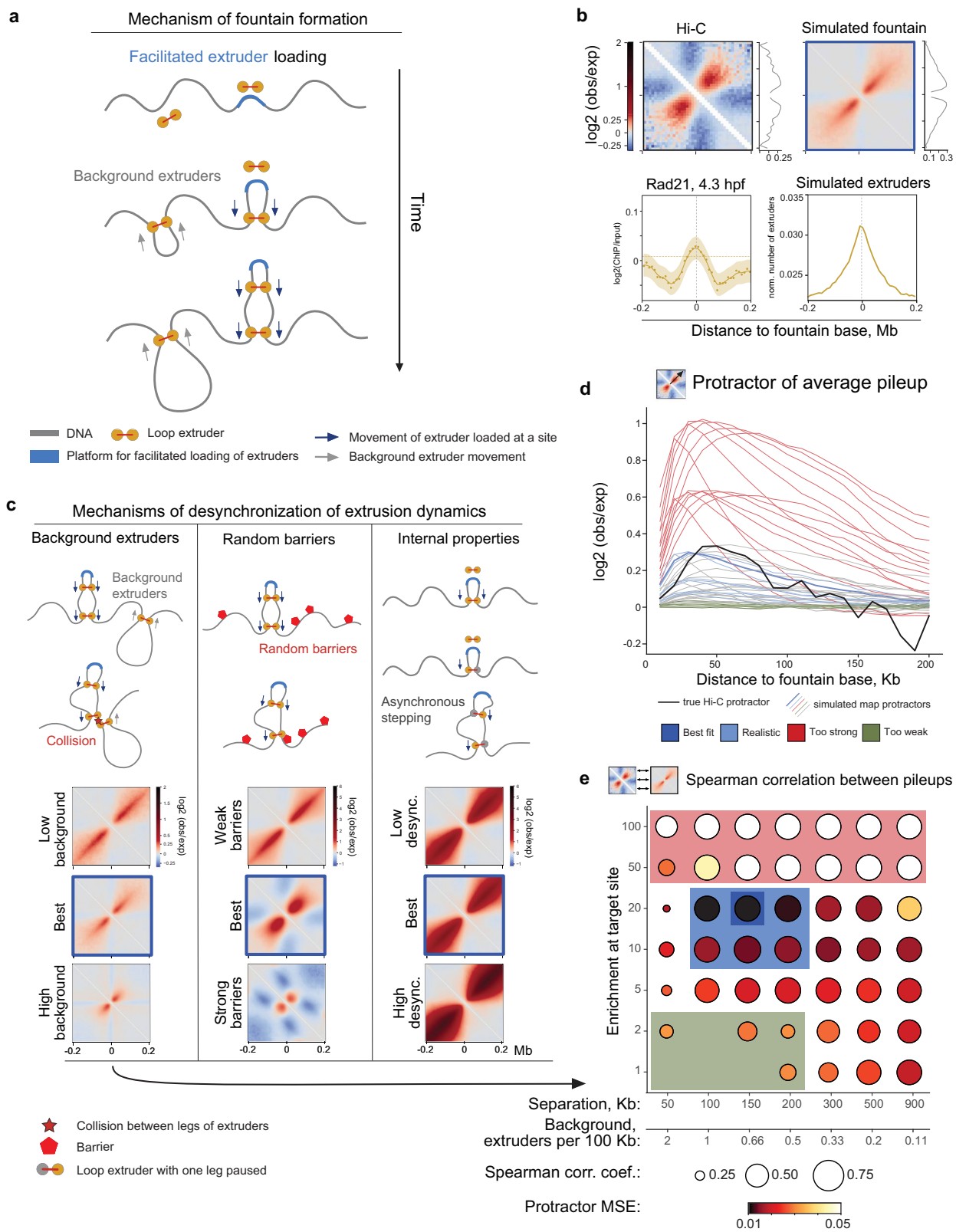

**a** Mechanism of fountain formation

Facilitated extruder loading

Background extruders

Time

DNA | Loop extruder
Platform for facilitated loading of extruders
Movement of extruder loaded at a site
Background extruder movement

**c** Mechanisms of desynchronization of extrusion dynamics

Background extruders | Random barriers | Internal properties

Background extruders
Collision
Random barriers
Asynchronous stepping

Low background | Weak barriers | Low desync.
Best | Best | Best
High background | Strong barriers | High desync.

-0.2 0 0.2 | -0.2 0 0.2 | -0.2 0 0.2 Mb

★ Collision between legs of extruders
⬠ Barrier
Loop extruder with one leg paused

**b** Hi-C | Simulated fountain
log2 (obs/exp)

Rad21, 4.3 hpf | Simulated extruders
log2(ChIP/input) | nom. number of extruders

Distance to fountain base, Mb

**d** Protractor of average pileup
log2 (obs/exp)
Distance to fountain base, Kb
— true Hi-C protractor | ⫽ simulated map protractors
Best fit | Realistic | Too strong | Too weak

**e** Spearman correlation between pileups
Enrichment at target site

Separation, Kb: 50 100 150 200 300 500 900
Background, extruders per 100 Kb: 2 1 0.66 0.5 0.33 0.2 0.11
Spearman corr. coef.: ○ 0.25 ◯ 0.50 ◯ 0.75
Protractor MSE: 0.01 0.05

structures formed by extrusion are different from those seen in adult cells, such as TADs and dots. We discovered a distinct class of chromosome structures–fountains, with more than a thousand emerging across the genome around the time of genome activation. We demonstrated that fountains are distinct from known features of local chromosome folding, like TADs, dots, and stripes–all of which are linked to the pausing of extruding cohesin by CTCFs[12], and to a lesser extent by transcribed genes[48]. Fountains, in contrast, are not associated with CTCFs or positions of TAD borders, and do not directly depend on transcription. Importantly, we demonstrated that fountains are not just a peculiarity of a Hi-C map, but rather, they show a striking association with developmental enhancers and a causal relationship with them. In synergy with our efforts, studies of Lüthi et al.[30] and Kim et al.[31] found enhancer-associated fountains (jets) in *C. elegans*, the

**Fig. 5 | Facilitated extruder loading as a mechanism of fountain formation.** Source data are provided as a Source Data file. **a** Proposed mechanism of fountain formation. The extruder molecule (cohesin complex) preferentially loads at the target region (highlighted in blue) and begins extruding DNA in two directions. Multiple extruders can load at the target sites, forming a hairpin-like structure. Extruder molecules can also load at other genomic loci (background extruders with lower loading rates). **b** Comparison of Hi-C average fountain to the best fit of the model with facilitated loading and background cohesins. (top left) Average pileup of fountains in experimental Hi-C data, with the protractor on the right. (bottom left) Distribution of cohesin subunit Rad21 at 4.3 hpf[4] around fountains, display elements as in Fig. 2a, b. (top right) Average simulated Hi-C obtained with the best parameters according to (**d**, **e**), Supplementary Methods "Goodness of fit of the simulations to real data". (bottom right) Cohesin coverage profile in simulations, total number of extruders per monomer, normalized by the simulation size. **c** Three mechanisms of desynchronization of extrusion dynamics: (left) cohesin movement is desynchronized due to collisions with background cohesins, (center) cohesin sides are stopped by random barriers, (right) internal properties of cohesin lead to asynchronous stepping of two sides. All three scenarios lead to the emergence of fountains with increasing spread with distance from the loading site. **d, e** Parameter sweep of the model of facilitated cohesin loading in the presence of background cohesin loading (c, left). **d** Intensity of the fountain measured by protractor for average fountain from Hi-C (black) and various simulation parameters: red – fountains are too strong, green–fountains are too weak, blue – fountains are in a similar range to the Hi-C data. Protractors for the same parameters as in (**e**). **e** Similarity of the simulated fountains to the real Hi-C data. Each circle represents a comparison of the simulation round to the experimental Hi-C fountain. Color represents the mean square error (MSE) of the protractor. Circle size represents the Spearman correlation of the fountain shape. Boxes of different colors represent various types of simulation outcomes.

organism that lacks CTCF, supporting our finding that fountains are formed on enhancers by a CTCF-independent mechanism.

The association between fountains and enhancers is two-way specific: among ten functional annotation types, active enhancers were the only ones enriched at fountain locations. Conversely, an average Hi-C map at enhancers shows a pronounced fountain pattern. Importantly, the latter observation did not rely on definitions or calling of the fountains.

Knockouts of key pioneer transcription factors establish a causal link between early active enhancers, chromatin accessibility, and fountains. Knockouts of individual or combinations of transcription factors result in the loss of chromatin accessibility and enhancer activity at specific locations, leading to the corresponding loss of fountains at those sites. Interestingly, only 19% of fountains are affected by the loss of all three pioneer transcription factors, suggesting that other genomic elements may also contribute to their formation.

Our analysis suggests that fountains also form at latent enhancers, which become active only later in development and in specific tissues. In zebrafish, regulatory elements show little overlap between tissues and developmental stages[41,47]. These late and tissue-specific enhancers do not carry specific enhancer marks at 5.3 hpf and cannot be readily discerned. However, we found that fountains unaffected by the loss of pioneer transcription factors are enriched in enhancers that become active after gastrulation and in adult tissues, and show a broad association with developmental genes. Altogether, this suggests that fountains formed upon ZGA form on active as well as on latent enhancers.

Which factors enable fountain formation at active and latent enhancers? The most parsimonious explanation is chromatin opening by pioneer factors. Many of such factors are already present in the embryo at sufficiently early stages, e.g., zebrafish orthologs of mouse pioneer transcription factors Eomesodermin and Brachyury[61,62]. These factors can establish accessible chromatin at latent enhancers, thus triggering fountain formation, way before these enhancers are fully activated. While enhancer activity is necessary for fountain formation, fountains are not essential for the formation of enhancers – a conclusion of a recent study in adult *C. elegans*[30] where loss of fountains upon cohesin COH-1 cleavage did not result in enhancer loss. Indeed, some histone marks and accessibility are mitotically inherited[63,64], while others are quickly established in G1, allowing rapid fountain formation in early G1 as we observe here (Fig. 6c).

Several lines of evidence point to association of fountains with cohesin. First, we found accumulation of cohesin at fountains, with the higher levels of cohesin associated with stronger fountains. Similarly, enhancers with higher levels of cohesin show a stronger fountain pattern. Second, fountains at enhancers in mouse cells vanish upon acute cohesin (Rad21) depletion, while getting much stronger upon depletion of Wapl, when cohesin processivity and density are increased. Furthermore, the loss of fountains in mitotic cells and their

gradual reemergence in G1 upon cohesin reloading are consistent with the cohesin-mediated mechanism (Fig. 6c). Consistently, a double knockout of CTCF and Wapl in mice reported the formation of plumes at open chromatin regions[29]. A recent study in *C. elegans* further shows fountain loss upon acute cohesin depletion[30,31]. Another recent study of partial depletion of cohesin in vivo, in postmitotic mouse dendritic cells, reported weakening of fountains that is accompanied by a drastic loss of immune function of these cells[20]. Finally, our simulations and those of Guo et al.[28] demonstrate that facilitated loading of cohesin followed by bidirectional extrusion leads to the formation of the fountain pattern.

Importantly, our simulations also reveal that (i) about -10-fold facilitation is sufficient, and that (ii) such facilitated loading needs to be accompanied by mechanisms that desynchronize reeling of DNA into an extruded loop from the two sides of cohesin. Such desynchronization can be an inherent property of cohesins, as was seen in vitro[50], or can result from collisions with randomly positioned barriers and/or with other extruding cohesins elsewhere.

While it is natural to expect cohesin loading at accessible regions, such as enhancers, surprisingly, other accessible regions, such as promoters, do not show any association with fountains. In fact, active promoters show an anti-fountain insulation pattern, consistent with the role of transcription as an extrusion barrier[48]. Given the overall similarity between enhancers and promoters, the selective loading of cohesin on enhancers is surprising. We imagine two mechanisms that could lead to the formation of fountains selectively on enhancers. The first mechanism would rely on selective recruitment of cohesin components to enhancers (e.g., by pioneer transcription factors), as have been reported for mouse ES cells[55–57], human cancer cell lines[65], mouse liver[66], and cultured *Drosophila* cells[67,68]. The second mechanism would rely on loading of cohesin at accessible regions–both at enhancers and promoters–but require a factor (e.g., an RNA polymerase[48]) that prevents cohesin's extrusion from promoters.

Thus, fountains provide evidence of facilitated loading of loop-extruding cohesins at enhancers. While individual fountains are detectable only in specific biological contexts and systems, it's likely that the underlying process of facilitated loading of cohesin at enhancers proceeds through the interphase and a broad range of conditions. Indeed, individual fountains are detectable in zebrafish embryos; they are hard to discern in adult cells (Fig. 1c). Similarly, individual fountains we detected in the early M-to-G1 transition (Fig. 6c, and Supplementary Fig. 7), become obscured by TADs, stripes, and dots (Fig. 6d, and Supplementary Fig. 7). Consistently, individual fountains have been reported in mouse quiescent thymocytes[28], or upon double Wapl/CTCF depletion[29], or in fungi upon activation of secondary metabolic gene clusters[32]. Yet pileups at enhancers in mouse ES cells (Fig. 6a, b), where individual fountains are not readily detectable, still provide evidence of facilitated loading of cohesin, becoming larger upon WaplKO and longer upon CTCF depletion

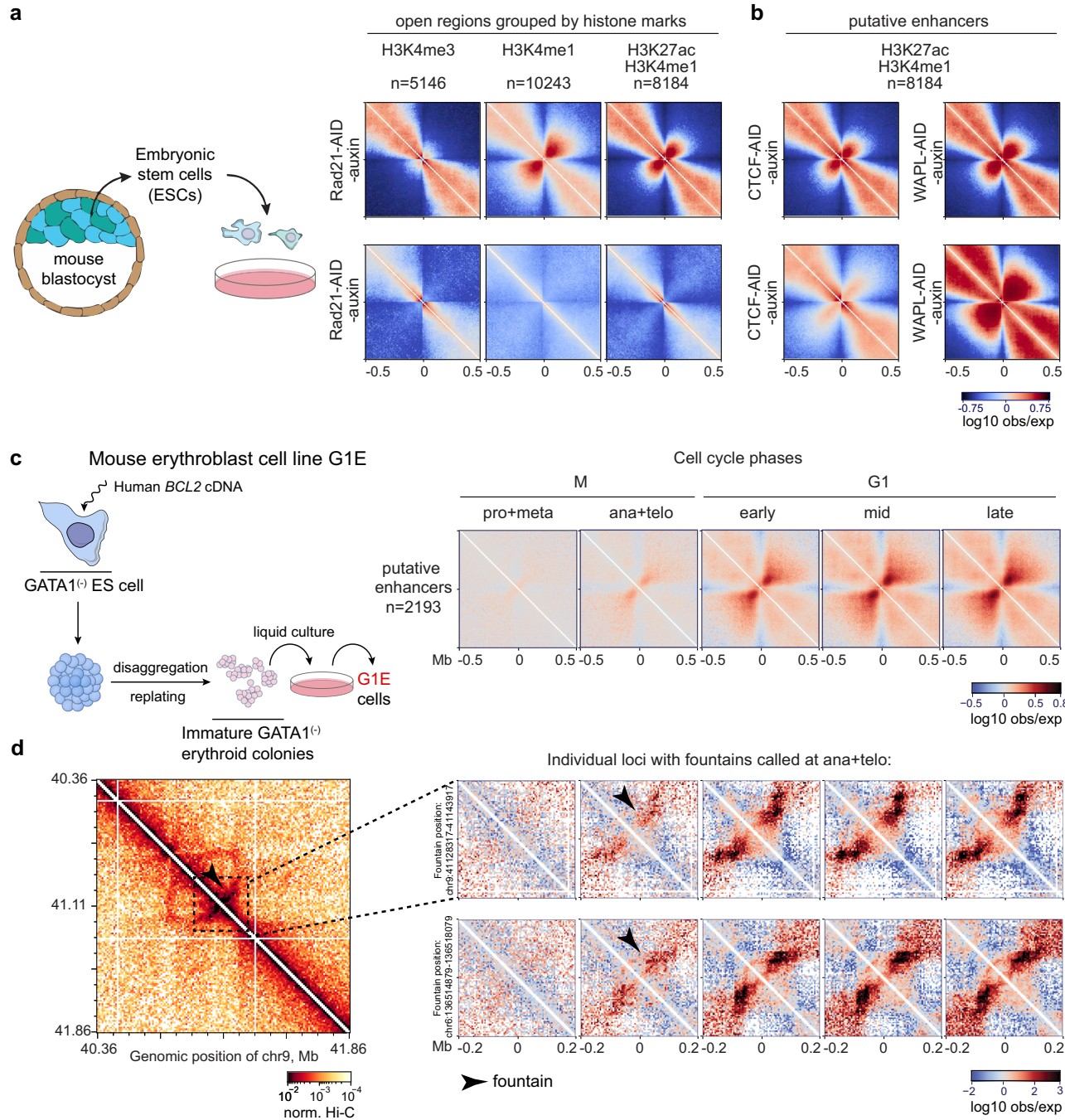

**Fig. 6 | Cohesin-dependent fountain-like structures emerge at the enhancers in mouse cell lines.** Source data are provided as a Source Data file. **a, b** Average Micro-C pileups at the indicated genomic elements in mouse embryonic stem cells (ESCs) in two conditions. Top: control condition (no auxin), bottom: auxin-inducible acute knockout of the indicated gene (Micro-C from[58]). Open chromatin regions were assessed by DNase sites from[86]. Histone modification ChIP-seqs from the same study. Overlaps with CTCF peaks were excluded from the analysis (CTCF ChIP-seq data from[58], see Methods "Hi-C snipping and average pileup"). Micro-C bin size 5 Kb. **a** Fountains form on enhancers and depend on cohesin in mouse ESCs. Left: Embryonic origin of mouse ESCs. Right: Pile up on non-CTCF open chromatin regions (DNase I peaks) grouped by enrichment on indicated chromatin marks for mouse embryonic stem cells E14 (ENCODE data[87]) in control and upon Rad21 acute degradation. Note that fountain-like structures form on enhancers but not

promoters (compare to Fig. 2d). Micro-C bin size 5 Kb. **b** Pile up of enhancers (defined as H3K4me1+ H3K27ac+ open chromatin regions, same as in Fig. 6a, right) in control and upon CTCF and WAPL acute degradation. **c, d** Fountain-like structures on non-CTCF potential enhancers form during G-phase and start to emerge as early as in ana-telophase. **c** Left: Origin of G1E erythroblast cell line; modified from[88]. Right: Pileup of active enhancers in G1E erythroblast cell line synchronized in the indicated phases of the cell cycle (Hi-C data and ChIP-seq from[59]). Overlaps with CTCF peaks (with 10 Kb offset) were excluded from the analysis in all cases. **d** Examples of individual fountains formed at enhancers in G1E cells. Left: Hi-C map around the enhancer with the potential fountain at ana-telophase. Right: Hi-C maps of the enhancers with potential fountains in the exit from mitosis. Additional individual examples are shown in Supplementary Fig. 7. Black arrows indicate areas with visible fountains.

(Fig. 6b). Similarly, pileups at enhancers continue showing fountain signature in later G1 in erythroblastoid cells, and in more developed zebrafish (Fig. 1d).

If cohesin is loaded preferentially at enhancers, why are individual fountains seen only at specific times and conditions? It's likely that visibility of individual fountains requires unobscured extrusion by loaded cohesins. Indeed, collisions between facilitated-loaded cohesin and other cohesins or polymerases—as well as stalling and sequestration at CTCFs—transform fountains and overshadow them with TADs and other structural features. Consistent with this mechanism, we find that at early developmental stages, cohesin accumulates at fountains rather than at CTCF sites, whereas at later stages, cohesin redistributes from fountain bases to CTCF sites and TAD borders (Supplementary Fig. 2c).

The continued detection of fountain signatures in pileups at enhancers in later developmental stages and during later interphase indicates that the facilitated loading process remains active, while individual fountains are less discernible (Fig. 6d). Altogether, these observations suggest that fountains are most prominent in early development and early G1, eventually evolving into well-recognized extrusion-mediated structures.

What is the functional role of facilitated loading of cohesin at enhancers? Paradoxically, fountains facilitate contacts between regions flanking an enhancer, but do not facilitate contacts of the enhancer itself. This prompts the question: how can fountains help enhancers to perform their functions? One possibility is that facilitated cohesin loading at enhancers facilitates contacts between other regulatory regions on one side of an enhancer with genes on the other, thus helping other nearby enhancers. Another possible role of enhancer-loaded cohesin can be to stall at the enhancer, thus performing a one-sided extrusion, and thus facilitate formation of its interactions with distal genomic elements. Such loading of cohesins at enhancers may be part of the mechanism by which cohesin's loop extrusion facilitates enhancer-promoter interactions[16–20]. If stalling at enhancers and one-sided extrusion events are sufficiently infrequent, they might not leave a noticeable signature in the fountain pattern, yet facilitate contacts with target genes. Ultimately, facilitated loading of cohesin at enhancers constitutes a promising avenue for future investigation into the principles governing enhancer–promoter communication.

Importantly, our simulations suggest an ~10-fold higher loading rate of cohesins at enhancers, compared to non-enhancer chromatin. Yet, since enhancers constitute a minuscule fraction of genomic length (~0.7% at the dome stage in zebrafish based on[41]), the vast majority of cohesin is loaded at non-enhancer chromatin.

Broadly, our analysis shows that loop extrusion activity initiates upon ZGA. Yet structures formed by extrusion change through development due to different activity of cohesin-regulating factors. Cohesin-loading activity appears to be the primary factor mediating chromosome folding upon ZGA, while cohesin-stopping factors shape chromosomes at later stages. Surprisingly, key regulatory sites driving transcription, early enhancers, also appear to be the elements driving early folding events by recruiting cohesins. Extruding from enhancer, cohesins form the distinct elements of chromosome architecture – fountains. While fountains are best seen in early development, it's likely that the underlying facilitated loading of cohesin at enhancers proceeds through development. This role of enhancers in actively folding chromosomes can shed light on the mechanisms of long-range regulatory activity of enhancers in development and beyond.

## Methods
### Experimental model and subject details
All experiments were performed in accordance with German Animal Protection Law (TierSchG) and European Convention on the Protection of Vertebrate Animals Used for Experimental and Other Scientific Purposes (Strasbourg, 1986[69]). The generation of double and triple mutants was approved by the Ethics Committee for Animal Research of the Koltzov Institute of Developmental Biology RAS, protocol 26 from 14.02.2019.

Wild-type fish of AB/TL and mutant strains were raised, maintained, and crossed under standard conditions as described by Westerfield[70]. Embryos were obtained by natural crossing (4 males and 4 females in 1.7 l breeding tanks, Techniplast). Wild-type and mutant embryos from natural crosses were collected in parallel in 10–15 min intervals and raised in egg water at 28.5 °C until the desired stage. The staging was performed following the Kimmel staging series[34]. Stages of the mutant embryos were indirectly determined by observation of wild-type embryos born at the same time and incubated under identical conditions.

### Generation of MZ*sn*, MZ*pn* double mutant and MZ*triple* triple mutant embryos and maintenance of the mutant fish lines
Maternal-Zygotic (MZ) homozygous mutant embryos MZ*sn* (MZ*sox19b*[m1434]*nanog*[m1435]) were obtained in three subsequent crossings. First, MZ*nanog*[m143544] homozygous null-mutant males were crossed with MZ*sox19b*[m1434 42] homozygous null-mutant females. The double heterozygous fish were raised to sexual maturity and incrossed. The progeny developed into phenotypically normal and fertile adults. Genomic DNA from tail fin biopsies was isolated and used for genotyping (see Supplementary Methods).

MZ*sn* homozygous mutant embryos were obtained in three subsequent crossings. First, MZ*nanog*[m1435] homozygous males[44] were crossed with MZ*spg*[m793 43] homozygous females. *Spg*[m793] allele carries an A- > G point mutation in the splice acceptor site of the first intron of Pou5f3 gene, which results in the frameshift starting at the beginning of the second exon, before the DNA-binding domain. Spg[m793] is considered to be a null allele. The double heterozygous fish were raised to sexual maturity and incrossed. To bypass the early requirement for Pou5f3 in the *spg*[m793] homozygous mutants, one-cell stage embryos were microinjected with 50–100 pg synthetic Pou5f3 mRNA as previously described[43]. The fish were raised to sexual maturity (3 months), and genomic DNA from tail fin biopsies was isolated and used for genotyping (see Supplementary Methods). The line was maintained by crossing *spg*-/-, *nanog* -/- males with *spg*-/-, *nanog* +/- females and microinjecting Pou5f3 mRNA in each generation.

To obtain the triple Maternal-Zygotic homozygous mutant embryos MZ*triple*, MZ*sn* double homozygous males were crossed with MZ*ps* (MZ*sox19b*[m1434]*spg*[m793] double homozygous females[42]. The *sox19b*-/-; *spg* + /-; *nanog* +/- progeny was raised to sexual maturity and incrossed. The progeny was microinjected with 50–100 pg synthetic Pou5f3 mRNA at one cell stage, raised to sexual maturity and genotyped (see Supplementary Methods). MZ*triple* embryos for experiments were obtained from incrosses of triple homozygous fish. The line was maintained by crossing *sox19b*-/-;*spg*-/-, *nanog* -/- males with *sox19b*-/-;*spg*-/-, *nanog* +/- females and microinjecting Pou5f3 mRNA in each generation.

### Zebrafish sperm collection
Adult male fish were anesthetized with Tricaine (4% 3-Aminobenzoic acid ethyl ester, pH 6.7) and then positioned with the anal area above. The sperm was taken using a capillary, mixed with 5 µl E400 buffer (9.7 g KCl, 2.92 g NaCl, 0.29 g CaCl$_2$ −2H$_2$O, 0.25 g MgSO$_4$ −7H$_2$O, 1.8 g D-(+)-Glucose, 7.15 g HEPES in 1 L dH$_2$O, pH 7.9) and then with 150 µl SS300 buffer (0.37 g KCl, 8.2 g NaCl, 0.15 g CaCl$_2$ −2H$_2$O, 0.25 g MgSO$_4$ −7H$_2$O, 1.8 g D-(+)-Glucose, 20 mL of 1 M Tris-Cl, pH 8.0, in 1 L dH$_2$O). The collected sperm was used for nuclei isolation.

### Hi-C library preparation
The embryos were obtained from natural crossings in mass-crossing cages (4 males + 4 females). 5-10 cages were set up per genotype, and the eggs from different cages were pooled. The freshly laid eggs were

collected in 10-15-minute intervals. Embryos were incubated at 28.5 °C and dechorionated with pronase E (0.3 mg/ml) shortly before the desired stage. 400-600 embryos were homogenized in 2 ml 0.5 % Danieau's with protease inhibitor cocktail (PIC, Roche) and 1 % (v/v) Methanol-free Formaldehyde (Pierce) and fixed for 10 min on a rotating platform. The fixation was stopped with 0.125 M Glycine by shaking for 5 min on a rotating platform. Cells were pelleted on the tabletop centrifuge for 5 min, 500 g, and washed three times with PBST (16 mM $Na_2HPO_4$, 4 mM $NaH_2PO_4$, 0.08% NaCl(w/v), 0.002% KCl (w/v), 0.1% Tween 20, pH 7.5), with protease inhibitors. The cells were lysed for 1 min in 1 ml lysis buffer (10 mM Tris-HCl (pH 7.5), 10 mM NaCl, 0.5 % NP-40) on ice. The pellet was washed 2 times with 1 ml ice-cold 1x PBST. In order to count the obtained nuclei, the pellet was resolved in 1 ml ice-cold 1x PBST, of which 10 μl were diluted 1:1 with 12 μM Sytox® green. The nuclei were scored under a fluorescence microscope using the Neubauer counting chamber. The residual nuclei were snap-frozen in liquid nitrogen and stored at −80 °C. 2.5-3 million nuclei were used for one Hi-C experiment according to the published protocol[71]. For Hi-C data processing, see Supplementary Methods "Hi-C data mapping".

## Fountain calling with *fontanka*

*Fontanka* is a *cooltools*[38]-based tool for calling patterns in Hi-C maps (Supplementary Fig. 1e) that we used to search for fountains in the Hi-C maps (https://github.com/agalitsyna/fontanka). Fontanka takes the Hi-C map in .cool format and a reference fountain matrix as an input.

Reference fountains were manually marked in chromosomes 1 and 2. Average pileup of these fountains was collected with *cooltools*[38] and used as a reference for fountain calling.

Four steps of fountain calling with *fontanka* include:

(i) Snippets extraction. *Fontanka* rolls a square window along the main diagonal of the Hi-C matrix and extracts the observed over the expected signal. Each window is assigned to the genomic point of its center. We set the resolution to 10 Kb and the window size to 400 Kb.

(ii) Convolution. Instead of an insulation diamond window implemented in *cooltools* insulation[38], *fontanka* performs two types of convolution:

  (a) Fountain score calculation. Each snippet is convolved with the fountain mask, resulting in Pearson correlation as a measure for the fountain score for each genomic bin.

  (b) Noise score calculation. We applied another convolution round with the Scharr operator, approximating the gradient of the Hi-C map[72,73]. Too noisy or sharp patterns in Hi-C snippets result in high noise scores, while smooth Hi-C maps will have Scharr scores closer to zero.

(iii) Peak calling. *Fontanka* detects local maxima in the fountain score by the *cooltools find_peak_prominence* function[38]. This function detects peaks in the 1D genomic signal of the fountain score and assigns peak prominence to each one. The resulting peaks were considered candidate fountains (Supplementary Fig. 1e).

(iv) Thresholding and filtration. In order to minimize the contribution of false fountain calls, we select the most confident set of fountains (Supplementary Fig. 1e):

  (a) We first remove the fountains with weak prominence by applying Li's iterative method for finding the separation point by using the slope of the cross-entropy[74], as implemented in *scikit-learn*[75].

  (b) Next, we remove fountains that fall too close (<50 Kb) to the nearest bad bin.

  (c) We remove all the fountains that have a negative correlation with the fountain mask.

  (d) We next require that the fountains detected in the merged dataset are detected in individual replicates with an offset of 20 Kb (on either side from the fountain).

(e) We apply a filter for potential genomic rearrangements and misassembly problems based on fountain calls in replicates and at 2.75 hpf (see Supplementary Methods "Fountain calling with *fontanka*" for details).

(f) Another filter for potential genomic misassembly problems filters out the top 25% of candidate fountains based on their Scharr score.

The final list of fountains with fountain scores and fountain peak scores is provided as Supplementary Dataset 1. We fixed the software versions to fontanka v0.1, bioframe v0.4.1[76], cooler v0.8.11[77], and cooltools v0.5.2[38] in this work.

## Simulations of loop extrusion

Simulations of loop extrusion (Fig. 5, Supplementary Fig. 5–6) were based on the *OpenMM*-based[78] Python library *Polychrom*[79]. The implementation is available at (https://github.com/agalitsyna/polychrom_workbench) under the *targeted_extrusion* directory.

(i) 1D simulation. We simulated the translocation of extruders on a 1D lattice[80], where each point represented a genomic position. Each position can either (1) have a regular role or (2) serve as a loading platform for the targeted extruder. For regular positions, we assumed uniform probabilities of loading and unloading. For loading platforms, we set up the probability of cohesin landing increased by the enrichment factor. For the basic simulations of targeted loading, we assumed that there are (1) no boundaries that stall cohesin legs, (2) only other cohesins can stall the legs of cohesins, (3) legs of cohesin (if not stalled) are moving simultaneously, (4) free leg of cohesin with another stalled leg can move. For simulations with random barriers, we assumed that (1) cohesins load only at the loading platforms, (2) random barriers are placed at each genomic position in a 100 Kb window around the platform, (3) random barriers stall the cohesin leg with some probability. For simulations with decoupled legs of cohesin, we assumed that (1) cohesins load only at the loading platforms, (2) the probability of stepping is set for each leg independently.

(ii) 3D simulation. As in our previous works[12,39,81], we represented chromatin as a polymer with spherical monomers connected by harmonic bonds with stiffness and soft-core repulsive potential. Simulations were performed with variable Langevin integrator in the periodic boundary conditions[12] with volume density of 0.1. For simulations and setting up the bonds, we used the *OpenMM*-based[78] Python library *Polychrom*[79]. A harmonic bond connected the two monomers held by the extruder. The number of 3D-simulation time steps per 1D simulation step was set to 200[48].

(iii) In silico reconstruction of interaction probabilities. To calculate contact maps, we recorded all pairs of monomers within 5 monomer radii. We then averaged the interactions over multiple equivalent groups to obtain a simulated 2.5 Mb region with fountains. To obtain a simulated average fountain at 10 Kb resolution, we snipped 200 Kb windows around the loading platforms and coarse-grained them by a factor of 10. We then normalized the snippets by the expected.

(iv) Goodness of fit of the simulations to real data. The goodness of fit was assessed by the correlation of the simulated average fountain with the Hi-C average fountain. For additional details, see Supplementary Methods "Goodness of fit of the simulations to real data" section.

(v) Parameter sweep. To automate the parameter sweep, we implemented a *snakemake*[82] pipeline combining all simulation steps. Tested parameters and the resulting goodness of fit are presented in Fig. 5d (mean square error (MSE) of the protractor) and Supplementary Fig. 5b (Spearman correlation coefficient).

## Reporting summary

Further information on research design is available in the Nature Portfolio Reporting Summary linked to this article.

## Data availability

Raw and processed Hi-C data for zebrafish embryogenesis is available at GEO at (https://www.ncbi.nlm.nih.gov/geo/query/acc.cgi?acc=GSE195609). The list of zebrafish fountains is available in Supplementary Dataset 1, *Xenopus* and medaka fountains in Supplementary Dataset 2, zebrafish TAD boundaries in Supplementary Dataset 3, zebrafish initiation zones at 4.3 hpf in Supplementary Dataset 4, and fountains group assignment based on chromatin accessibility change in mutants is available in Supplementary Dataset 5. The detailed list of the sequencing datasets is available in Supplementary Dataset 6. Additional datasets (including intermediary processed files) are available at OSF at (https://osf.io/mt4vf). Interactive HiGlass views are available at Resgen at (https://resgen.io/galitsyna/Zebrafish_embryogenesis/). Source data are provided with this paper.

## Code availability

*Fontanka* tool for fountain calling (with examples) is available at (https://github.com/agalitsyna/fontanka) (v0.2, version available at https://doi.org/10.5281/zenodo.18018594). Simulations of facilitated loop extrusion are available at (https://github.com/agalitsyna/polychrom_workbench) under the directory targeted_extrusion. TAD calling for zebrafish is available at (https://github.com/encent/danio-2022). Code for reproducing the figures from the paper (main Figures and Supplementary Figs.) is available at (https://github.com/agalitsyna/embryonic-chromatin).

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

## Acknowledgements

We are grateful to Edward Banigan for proofreading the text and to all members of the Mirny lab for many productive discussions. We thank Damir Baranasic and Chris Sansam for sharing unpublished data. A.G. thanks Max Imakaev, Nezar Abdennur, and Anton Goloborodko for early ideas for the polymer simulations; Henrik Pinholt, Simon Grosse-Holz, and Emily Navarrete for ideas on the model and figures; Antoine Coulon, Mark Pownall, and Christopher Bohrer for the thorough discussions of the results and critical assessment of the hypotheses; students Alexey Shkolikov (FBB MSU) and Eli Rybnikova (PRIMES MIT) for their preliminary analysis of fountains. D.O. is grateful to Prof. Lev Yampolsky for the consultation about statistical analysis. This work was supported by DFG-ON86/5-1, DFG ON86/6-1 and DFG-EXC2189 for D.O., by NIH GM114190 to L.M., by RSF 23-14-00136 to M.S.G., and by RSF 21-64-00001-P to S.R.; A.G. and L.M. were also supported by NIH R01 AI164728 and NIH R01 NS113929.

## Author contributions

S.U.: Hi-C libraries; A.G.: Hi-C and genomic analysis, polymer simulations; M.B.: replication timing data analysis, N.B., K.P., M.S.G.: bioinformatics; S.U., M.V., M.G., D.O.: zebrafish mutants; D.O.: study design; S.R., D.O.: supervision of the wet part, funding acquisition; M.S.G., L.M.: supervision of Hi-C and genomic analysis and simulations. A.G. wrote the first draft, and S.U., S.R., L.M., and D.O. developed and edited the manuscript.

## Funding

## Competing interests

The authors declare no competing interests.

## Additional information

[1]Institute for Medical Engineering and Science, Massachusetts Institute of Technology, Cambridge, MA, USA. [2]Institute of Gene Biology, Russian Academy of Sciences, Moscow, Russia. [3]Department of Molecular Biology, Faculty of Biology, M.V. Lomonosov Moscow State University, Moscow, Russia. [4]Department of Developmental Biology, University of Freiburg, Freiburg, Germany. [5]Signaling Research Centres BIOSS and CIBSS, Freiburg, Germany. [6]Spemann Graduate School of Biology and Medicine (SGBM), Freiburg, Germany. [7]Independent researcher, Moscow, Russia. [8]Faculty of Bioengineering and Bioinformatics, M.V. Lomonosov Moscow State University, Leninskiye Gory, 1, building 73, Moscow, Russia. [9]Independent researcher, P.O. Box 125476-22 Moscow, Russia. [10]Department of Physics, Massachusetts Institute of Technology, Cambridge, MA, USA. [11]Koltzov Institute of Developmental Biology RAS, Moscow, Russia. [12]Present address: Centro Nacional de Análisis Genómico (CNAG), Baldiri Reixac 4, Barcelona, Spain. [13]These authors contributed equally: Aleksandra Galitsyna, Sergey V. Ulianov. [14]These authors jointly supervised this work: Leonid Mirny, Daria Onichtchouk. ✉e-mail: galitsyn@mit.edu; leonid@mit.edu; daria.onichtchouk@biologie.uni-freiburg.de

