## [Transparent Peer Review file · Nature Communications]

Extrusion fountains are hallmarks of chromosome organization emerging upon zygotic genome activation

Corresponding Author: Professor Leonid Mirny

Version 0:

Reviewer comments:

Reviewer #1

(Remarks to the Author)

The authors have done a good job in addressing the concerns raised in my previous reviews of their earlier submissions - I have no further concerns.

(Remarks on code availability)

Reviewer #2

(Remarks to the Author)

I thank the authors for engaging with my comments on their manuscript

The strength of this manuscript by Galitsyna et al. remains the description of 'fountains' that arise transiently around the time of zebrafish zygotic genome activation where they selectively emanate from active enhancers. Selective formation of fountains at active enhancers is interesting because there are no other known enhancer-specific 3D features. Fountains are linked to enhancer-like chromatin state, and strongly associated with H3K3me1 and H3K27ac. The pioneer transcription factors Pou5f3, Sox19b, and Nanog (PSN) are required for the establishment of developmental enhancers, and consequently for the emergence of enhancer-associated fountains.

As repeatedly discussed in previous rounds of review, the authors' claim that fountains also mark enhancers in ES cells is weak because it is based solely on aggregate analysis, and the authors cannot visualise individual fountains. This limitation also weakens the authors' assertion that the formation of fountains relies on cohesin.

The field has moved on since the original submission of the manuscript by Galitsyna et al. and cohesin-dependent fountain formation is now well described in various model organisms, as is the formation of strong, individually visible fountains at enhancers in mouse ES cells depleted of CTCF and WAPL (the current NAR version of Liu et al. should be cited <https://doi.org/10.1093/nar/gkaf549>).

My suggestion for the way forward is that the authors focus on the strengths of their own findings, which are sufficiently interesting to warrant publication. The issue of cohesin-dependence could be dealt with in passing. For example, in the 'Cohesin accumulation is associated with fountain strength' section the authors could simply reference published papers that show cohesin-dependence of fountain-like structures in worms, mouse ES cells and mouse lymphocytes.

The authors could then minimize the presentation of ES cell and G1E pileups and instead discuss their findings in the light of the published literature.

Note that the authors' description of Extended Data Figure 7 (...fountains appear at enhancers in the mouse cell cycle as

early as anaphase/telophase (Fig. 6c, ED Fig. 7), but become increasingly difficult to distinguish from dots and stripes at later stages of mitotic exit) contradicts the pileups shown in Fig 6c, where fountain-like pileup patterns strengthen as G1 progresses. This appears inconsistent with the authors' narrative of fountain erasure by the formation of other Hi-C contacts, and also question whether individual fountains behave as the pileups would suggest.

(Remarks on code availability)

Reviewer #1 (Remarks to the Author):

The authors have done a good job in addressing the concerns raised in my previous reviews of their earlier submissions - I have no further concerns.

Reviewer #2 (Remarks to the Author):

I thank the authors for engaging with my comments on their manuscript

The strength of this manuscript by Galitsyna et al. remains the description of 'fountains' that arise transiently around the time of zebrafish zygotic genome activation where they selectively emanate from active enhancers. Selective formation of fountains at active enhancers is interesting because there are no other known enhancer-specific 3D features. Fountains are linked to enhancer-like chromatin state, and strongly associated with H3K3me1 and H3K27ac. The pioneer transcription factors Pou5f3, Sox19b, and Nanog (PSN) are required for the establishment of developmental enhancers, and consequently for the emergence of enhancer-associated fountains.

As repeatedly discussed in previous rounds of review, the authors' claim that fountains also mark enhancers in ES cells is weak because it is based solely on aggregate analysis, and the authors cannot visualise individual fountains. This limitation also weakens the authors' assertion that the formation of fountains relies on cohesin.

Response: We revised the manuscript by highlighting that pileups reflect the continued process of loading at enhancers, during development and through the cell cycle, while individual fountains evident in early G1 or in early development become harder to discern.

The field has moved on since the original submission of the manuscript by Galitsyna et al. and cohesin-dependent fountain formation is now well described in various model organisms, as is the formation of strong, individually visible fountains at enhancers in mouse ES cells depleted of CTCF and WAPL (the current NAR version of Liu et al. should be cited <https://doi.org/10.1093/nar/gkaf549>).

In the revised manuscript, we extensively cite other recent observations of fountains, many of which used our algorithm to detect them. Among them, we also mention a recent detection of individual fountains Wapl/CTCF depleted mouse ES cells, and disappearance of fountains upon additional cohesin depletion. These

observations further support our findings of cohesin-dependent “average” fountain at enhancers in wild-type mouse ES cells.

My suggestion for the way forward is that the authors focus on the strengths of their own findings, which are sufficiently interesting to warrant publication. The issue of cohesin-dependence could be dealt with in passing. For example, in the 'Cohesin accumulation is associated with fountain strength' section the authors could simply reference published papers that show cohesin-dependence of fountain-like structures in worms, mouse ES cells and mouse lymphocytes.

We cite all these works in the revised manuscript.

The authors could then minimize the presentation of ES cell and G1E pileups and instead discuss their findings in the light of the published literature.

We see the value in pileups as they suggest continued facilitation of cohesin loading at enhancers through the cell cycle.

Note that the authors' description of Extended Data Figure 7 (...fountains appear at enhancers in the mouse cell cycle as early as anaphase/telophase (Fig. 6c, ED Fig. 7), but become increasingly difficult to distinguish from dots and stripes at later stages of mitotic exit) contradicts the pileups shown in Fig 6c, where fountain-like pileup patterns strengthen as G1 progresses. This appears inconsistent with the authors' narrative of fountain erasure by the formation of other Hi-C contacts, and also question whether individual fountains behave as the pileups would suggest.

We revised the text to point out the lack of contradiction between ED Fig 7 and Fig 6c. Since fountains are mere reflections of the process of facilitated loading, pileup patterns remain strong through G1 progression, even when individual fountains are obscured by extrusion barriers. We extended Fig 6 to show changes in individual fountains through cell cycle.